# Child anemia in Cambodia: A descriptive analysis of temporal and geospatial trends and logistic regression-based examination of factors associated with anemia in children

Samnang Um [1]*, Michael R. Cope[2], Jonathan A. Muir[3]

**1** The National Institute of Public Health, Tuol Kork District, Phnom Penh, Cambodia, **2** Department of Sociology, Brigham Young University, Provo, Utah, United States of America, **3** The Global Health Institute, Emory University, Atlanta, Georgia, United States of America

* umsamnang56@gmail.com

**Data Availability Statement:** Our study used 2005, 2010, 2014 and 2021-22 Cambodia Demographic and Health Survey (CDHS) datasets. The DHS data

## Abstract

Anemia in children remains a public health concern in many resource-limited countries. To better understand child anemia in Cambodia, we examined temporal and geospatial trends of childhood anemia and used logistic regression to analyze its association with individual and household characteristics using data from the Cambodia Demographic and Health Surveys for 2005, 2010, and 2014. The prevalence of childhood anemia decreased from 62.2% in 2005 to 56.6% in 2014. The prevalence of childhood anemia was highest in Pursat (84.3%) for 2005, Kampong Thom (67%) for 2010, and Preah Vihear and Steung Treng (68.6%) for 2014. After adjusting for other variables, factors positively associated with childhood anemia included having a mother who was anemic (adjusted odds ratio (AOR) = 1.77, 95% CI: 1.58–1.97); being male vs. female (AOR = 1.20, 95% CI: 1.07–1.33), underweight (AOR = 1.24, 95% CI: 1.14–1.57), or stunted (AOR = 1.24, 95% CI: 1.09–1.41); or having had a recent episode of fever (AOR = 1.16, 95% CI: 1.03–1.31). Children were less likely to have anemia if they were older than 12 months. They were also less likely to have anemia if they were from a wealthier household (AOR = 0.64; 95% CI: 0.50–0.84) or had taken medications for intestinal parasites (AOR = 0.86; 95% CI: 0.89–0.93). These associations were generally consistent across time and space. Public health interventions and policies to alleviate anemia should be prioritized to address these factors across geospatial divides. Anemia remains highly prevalent among children aged 6–59 months in Cambodia.

## Introduction

Anemia is a serious public health problem worldwide [1]. According to World Health Organization (WHO) criteria, children aged 6–59 months are considered anemic if their hemoglobin level is below 11.0 grams per liter (g/dL), adjusted for altitude and smoking [2]. Anemia in children is a major cause of adverse health consequences such as stunted growth, impaired cognitive development, compromised immunity, disability and increased risk of morbidity and mortality [3]. Globally, anemia affected 40% (269 million) of children aged 6–59 months

are publicly available from the website at (URL: https://www.dhsprogram.com/data/available-datasets.cfm). The Shapefiles for administrative boundaries in Cambodia are publicly accessible through DHS website at (URL: https://spatialdata.dhsprogram.com/covariates).

**Funding:** The authors received no specific funding for this work.

**Competing interests:** The authors have declared that no competing interests exist.

in 2019 [1]. About 43% of all children in low-income and middle-income countries were diagnosed with anemia in 2019 [1]. Moreover, a higher prevalence of anemia has occurred in the South Asian region, with 52% (87 million) of children presenting with anemia in 2019 [1]. An estimated 145,073 children die annually worldwide due to iron deficiency from anemia [4]. Similar to many low-income and middle-income nations, anemia is a serious public health problem; among children aged 6–59 months in Cambodia, the proportion of those with anemia slightly decreased from 55.2% to 49% between 2014, and 2019, respectively [5].

Anemia has significant consequences for human health, as well as for social and economic development. Previous studies found than children with very severe anemia were 4.3 times more likely to die compared to children without anemia [6]. The causes of anemia in developing countries are typically grouped into three broad categories: nutritional deficiencies, infectious diseases, and genetic hemoglobin disorders [7]. In many low- and middle-income economies, the prevalence of anemia varies by socioeconomic factors such as maternal education, household wealth status, maternal employment, and residence [8–12]. Some studies have demonstrated evidence of anemia associated with child age and sex [10, 11, 13]. Moreover, children who had experienced fever, recent diarrhea had a higher likelihood of having anemia [14, 15]. Serval studies found maternal anemia increased the odds of having anemia among children [14–16]. Evidence from an analysis of nationally representative DHS data of 47 countries showed that in over 60% of countries, children with off-premises water access had significantly increased odds of anemia; children exposed to surface water had higher odds of anemia in over a quarter of countries [17].

The prevalence of anemia in Cambodia has slowly decreased in children aged under five years during 2014–2019 [5]. To further reduce rates of anemia in Cambodia, an improved understanding of its geographical distribution and associated risk factors is required. An enhanced understanding will help identify populations at greater risk for anemia and prioritize geographic areas for targeted interventions. To our knowledge, there are no published peer-reviewed studies that assess social and demographic factors associated with anemia among children aged 6–59 months in Cambodia over time. One prior study on anemia patterns and risk factors utilized data from CDHS 2014 [15]. This study included all children aged 6–59 months and pooled DHS data in South and Southeast Asian countries. An additional study aimed to describe trends in the nutritional status of children under age five in Cambodia based on four CDHS surveys taken from 2000 to 2014 to assess the effect of inequality (child's age, child's sex, mother's education, place of residence and wealth index) on nutrition [18]. Given the paucity of scholarship addressing this health concern among Cambodian children aged 6–59 months, in this study, we aimed to assess temporal and geospatial trends of anemia among children aged 6–59 months in Cambodia and examine factors associated with anemia among children aged 6–59 months in Cambodia. Understanding these factors may further support policy development and programs with more effective strategies and interventions in Cambodia to reduce the prevalence of anemia and associated health risks such as infant and child mortality among children aged 6–59 months.

## Materials and methods

### Ethics statement

The original CDHS protocol was approved by the National Ethical Committee for Health Research, Ministry of Health of the Cambodia (Ref: 056 NECHR) and the Institutional Review Board (IRB) of ICF in Rockville, Maryland, USA. Moreover, for the protection of using the secondary data in this study, the research proposal was reviewed and provided ethical approval (Ref: 225 NECHR) by the National Ethics Committee for Health Research, Ministry of Health

of the Cambodia [19]. CDHS data are publicly accessible upon request through the DHS website at (https://dhsprogram.com/data/available-datasets.cfm) [20]. Details of the questionnaires, procedures, and data collection methods can be found elsewhere and through the DHS website [21–23]. Interviewers explained the objective of the survey to parents/guardians of each participant under 18 years of age, and written informed consent was obtained from parents/guardians before data collection. The study contains no individual identifiers that could affect the confidentiality of the participants, and the data were used for analysis purposes only.

## Data sources and procedures

We used existing children's data from the 2005, 2010, and 2014 CDHS. The CDHS is a nationally representative population-based household survey that is regularly conducted roughly every 5 years. The survey typically uses two-stage stratified cluster sampling to collect samples from all of the Cambodian provinces, which have been divided into sampling domains. In the first stage, clusters, or enumeration areas (EAs), that represent the entire country are randomly selected from the sampling frame using estimates of probability proportional to cluster size (PPS). The second stage then involves the systematic sampling of households listed in each cluster or EA. Details of CDHS survey reported have been described elsewhere [21–23].

Data about the maternal and children demographic, their behaviors, their access to health care, and their household characteristics, as well as the health status of children, were collected when the enumerators identified children in the selected households. Anemia testing was performed on women and men aged 15–49, as well as children aged 6–11 months when parents or caretakers consented to the test. Blood samples were drawn using a finger prick or a heel prick in the case of women and men aged 15–49 and children aged 6–59 months [24]. Hemoglobin analysis was done on site using a battery-operated portable HemoCue 201+analyzer, adjusted for altitude and smoking [24]. HemoCue 201+analyzer was recommended as the gold standard device for venous blood measurements and have noted its great predictability (98.7%) [25] and sensitivity (93%) [26]. From the total of 23,687 children who were under five years old, 19,646 were of 6–59 months of age. Data on the children's hemoglobin level were available; ultimately, data for 10,434 children 6–59 months were included in the final analysis (see **Fig 1**).

## Measures

Anemia status among children aged 6–59 months was the outcome variable. The original variable for anemia level in the CHDS was recorded as a categorical variable representing "hemoglobin level of 11.0 g/dL is defined as not anemic," "10.0–10.9 g/dL is mild anemia," "7.0–9.9 g/dL is moderate anemia," and "less than 7.0 g/dL is severe anemia" based on the WHO hemoglobin level cut-off points [2]. Then, the original variable was recoded into the binary variable Anemia, with mild, moderate, and severe anemia coded as Anemic = 1 and non-Anemic = 0.

Right-hand sided variables included individual-level (maternal and child) and household-level characteristics. **Maternal characteristics** included age (coded as 1 = 15–24 (reference), 2 = 25–34, and 3 = 35–49 years); education (coded as 0 = no education (reference), 1 = primary, and 2 = secondary or above); marital status (coded as 0 = not married which included separated, divorced, and widowed, and 1 = married, which included married and cohabitating); employment(coded as 0 = not working (reference), 1 = working which included self-employed, professional, clerical, technical, managerial, sales, and services; skilled and unskilled manual work; maternal anemia (0 = non anemic and 1 = anemic); and smoking (coded as 0 = non-smoker and 1 = smoker).

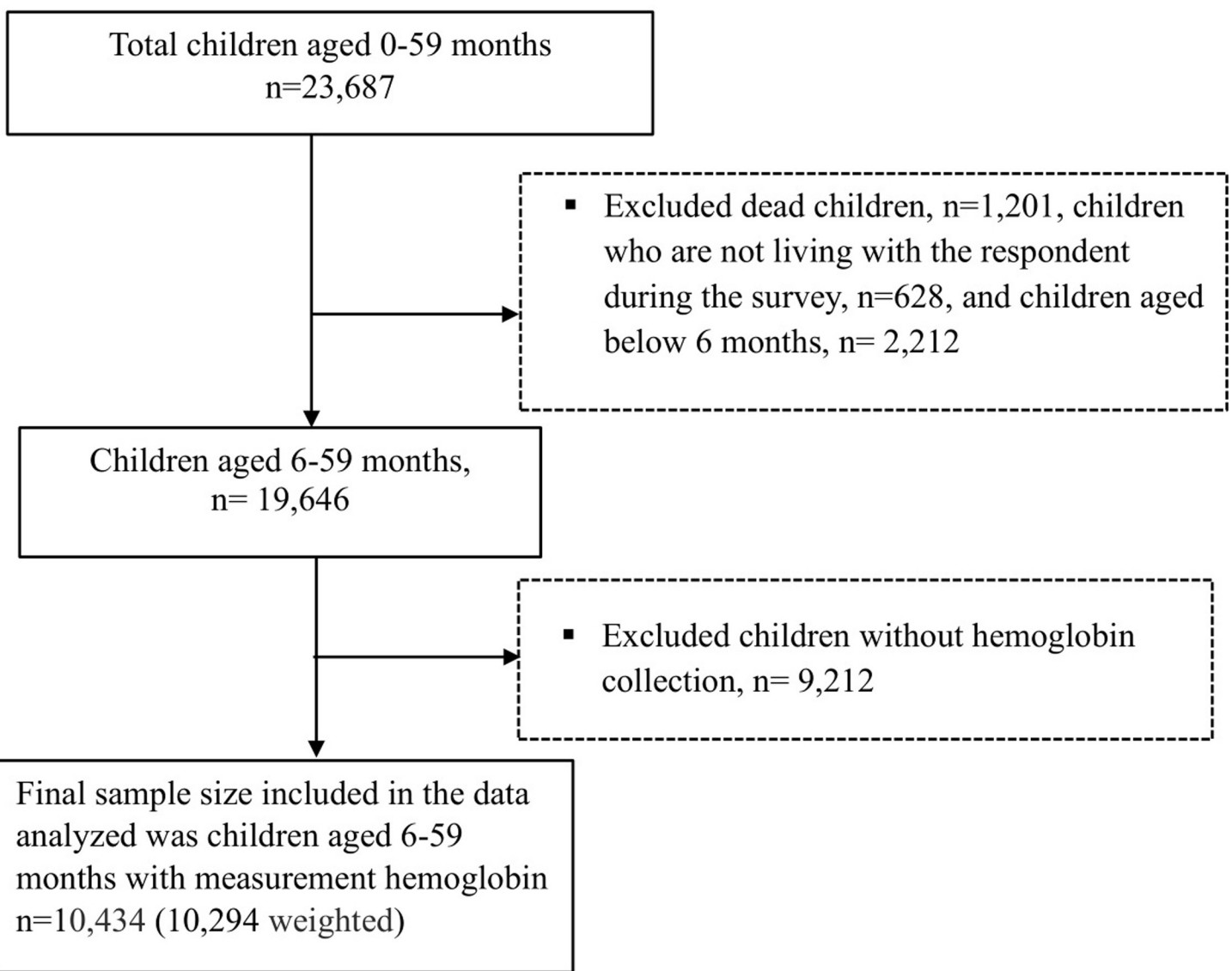

**Fig 1. Selection process and the final sample size from the 2005, 2010, and 2014 CDHS.**

**Child's characteristics** included sex (coded as 0 = female and 1 = male); age (coded as 1 = 6–11 (reference), 2 = 12–23, 3 = 24–35, 4 = 36–47, and 5 = 48–59 months); birth order (coded as 1 = 1–3 (reference), 2 = 4–5 and 3 = 6 or more); place of birth (coded as 1 = hospital (reference) which included public and private, 3 = home); stunting (coded as 0 = not stunted and 1 = stunted); underweight (coded as 0 = not underweight and 1 = underweight); and wasting (coded as 0 = not wasted and 1 = wasted); recent fever in past two weeks (coded as 0 = no and 1 = yes), recent diarrhea in past two weeks (coded as 0 = no and 1 = yes); Vitamin A supplementation within the last 6 months (coded as 0 = no and 1 = yes); intestinal parasite medication within the last 6 months (coded as 0 = no and 1 = yes); and Iron supplementation within the last 6 months (coded as 0 = no and 1 = yes).

**Household characteristics** included the DHS wealth index, coded as an ordinal level variable with richest = 1 (reference), richer = 2, middle = 3, poorer = 4, and poorest = 5 (we opted

to use the wealth index that was provided by each respective survey year of the CDHS as opposed to calculating a wealth index across the pooled data. This decision was based on prior research that found wealth status designation was comparable across the CDHS provided indices and an index based on data pooled across years) [23]; source of drinking water (coded as 1 = unimproved and 0 = improved), type of toilet (coded as 1 = unimproved and 0 = improved; and residence (coded as 1 = urban and 2 = rural). Cambodia's domains/provinces were regrouped for analytic purposes into a categorical variable with 4 geographical regions that were coded as plains = 1 (reference), Tonle Sap = 2, costal/sea = 3, and mountains = 4 (Plains included Phnom Penh, Kampong Cham/Tbong Khmum, Kandal, Prey Veng, Svay Rieng, and Takeo; Tonle Sap included Banteay Meanchey, Kampong Chhnang, Kampong Thom, Pursat, Siem Reap, Battambang, Pailin, and Otdar Meanchey; Coastal/sea included Kampot, Kep, Preah Sihanouk, and Koh Kong; and Mountains incnluded Kampong Speu, Kratie, Preah Vihear, Stung Treng, Mondul Kiri, and Ratanak Kiri). Survey year was coded as 1 = 2005 (reference), 2 = 2010, and 3 = 2014.

## Analytic strategy

Statistical analyses were performed using STATA version 17 (Stata Crop 2019, College Station, TX) [27]. We formally incorporated the DHS's complex sample design using the "survey" package; all estimations were carried out using the *svy* command in our descriptive and logistic regression analyses. Key maternal, child and household characteristics were described using weighted frequency distributions for specific survey years and for all years pooled together. A temporal series of maps illustrating provincial variations in the prevalence of anemia over time were created using ArcGIS software version 10.8 [28]. The underlying shapefiles of Cambodia provinces were obtained from the publicly accessible Spatial Data Repository associated with the DHS website (https://dhsprogram.com/data/available-datasets.cfm) [29]. Data quality assurance and minor data cleaning followed standard procedures in preparation for statistical analysis [30].

Cross-tabular weighted frequency distributions with Chi-square tests were used to assess the association between the variables of interest (including maternal, child household characteristics) and anemia status. Variables associated with anemia status with a p-value $\leq 0.05$ were used in the multivariate logistic regression analysis [31]. Women's age, residence, and survey years were included in the multivariate regression based on the literature and prior knowledge. Unadjusted logistic regression was used to analyze associations between anemia and maternal bio-demographic and household characteristics, geographical regions, and health-related factors. Results are reported as odds ratios (OR) with 95% confidence intervals (CI). Adjusted logistic regression was then used to assess factors associated with anemia after adjusting for other variables in the model. Results from the final adjusted model are reported as adjusted odds ratios (AOR) with 95% confidence intervals and corresponding p-values. Results from the final adjusted logistic regression model were considered statistically significant based on a p-value less than 0.05 and 95% confidence intervals that did not cross an AOR of 1.0. Potential multi-collinearity between right-hand sided variables was checked using a continuous version of the outcome variable and evaluating VIF scores after fitting an Ordinary Least Squares regression model (S1 Table).

## Results

A total of 10,294 children aged 6–59 months were included in the pooled data, there were 2,933 in 2005, 3,407 in 2010, and 3,955 in 2014, respectively. Overall, 49.2% of children were female, 23% had a mother who had at least completed secondary education, 46.5% had a

**Table 1. Descriptive statistics of individual and household socio-demographic characteristics.**

| Variables | 2005 (n = 2,933) | | 2010 (n = 3,407) | | 2014 (n = 3,955) | | 2005–2014 (n = 10,294) | |
|---|---|---|---|---|---|---|---|---|
| | Freq. | % | Freq. | % | Freq. | % | Freq. | % |
| **Maternal Characteristics** | | | | | | | | |
| **Age (years)** | | | | | | | | |
| 15–20 | 127 | 4.3 | 170 | 5.0 | 210 | 5.3 | 507 | 4.9 |
| 20–34 | 1,962 | 66.9 | 2,529 | 74.2 | 3,097 | 78.3 | 7,587 | 73.7 |
| 35–49 | 844 | 28.8 | 708 | 20.8 | 648 | 16.4 | 2,200 | 21.4 |
| **Education** | | | | | | | | |
| No education | 719 | 24.5 | 649 | 19.0 | 537 | 13.6 | 1,905 | 18.5 |
| Primary | 1,730 | 59.0 | 1,944 | 57.1 | 2,191 | 55.4 | 5,865 | 57.0 |
| Secondary plus | 484 | 16.5 | 814 | 23.9 | 1,226 | 31.0 | 2,365 | 23.0 |
| **Marital status** | | | | | | | | |
| Ever married | 132 | 4.5 | 155 | 4.5 | 179 | 4.5 | 466 | 4.5 |
| Married | 2,801 | 95.5 | 3,252 | 95.5 | 3,775 | 95.4 | 9,828 | 95.5 |
| **Employment** | | | | | | | | |
| Not working | 19 | 0.6 | 523 | 15.4 | 988 | 25.0 | 1,530 | 14.9 |
| Working | 2,914 | 99.4 | 2,884 | 84.6 | 2,966 | 75.0 | 8,764 | 85.1 |
| **Anemia status** | | | | | | | | |
| Non anemic | 1,438 | 49.0 | 1,898 | 55.7 | 2,151 | 54.4 | 5,488 | 53.3 |
| Anemic | 1,495 | 51.0 | 1,509 | 44.3 | 1,783 | 45.1 | 4,787 | 46.5 |
| **Smoking** | | | | | | | | |
| No-smoker | 2,767 | 94.3 | 3,300 | 96.9 | 3,842 | 97.1 | 9,908 | 96.3 |
| Smoker | 166 | 5.7 | 107 | 3.1 | 113 | 2.9 | 386 | 3.7 |
| **Child Characteristics** | | | | | | | | |
| **Place of birth** | | | | | | | | |
| Hospital | 565 | 19.3 | 1,736 | 51.0 | 3,300 | 83.4 | 5,601 | 54.4 |
| Home | 2,368 | 80.7 | 1,671 | 49.0 | 655 | 16.6 | 4,693 | 45.6 |
| **Age** | | | | | | | | |
| 6–11 | 324 | 11.0 | 409 | 12.0 | 455 | 11.5 | 1,188 | 11.5 |
| 12–23 | 696 | 23.7 | 786 | 23.1 | 959 | 24.2 | 2,440 | 23.7 |
| 24–35 | 620 | 21.1 | 763 | 22.4 | 874 | 22.1 | 2,257 | 21.9 |
| 36–47 | 673 | 22.9 | 710 | 20.8 | 852 | 21.5 | 2,236 | 21.7 |
| 48–59 | 621 | 21.2 | 739 | 21.7 | 815 | 20.6 | 2,174 | 21.1 |
| **Sex** | | | | | | | | |
| Female | 1,485 | 50.6 | 1,632 | 47.9 | 1,943 | 49.1 | 5,060 | 49.2 |
| Male | 1,448 | 49.4 | 1,775 | 52.1 | 2,011 | 50.8 | 5,234 | 50.8 |
| **Birth order** | | | | | | | | |
| 1–2 | 1,903 | 64.9 | 2,638 | 77.4 | 3,345 | 84.6 | 7,887 | 76.6 |
| 3–5 | 610 | 20.8 | 502 | 14.7 | 439 | 11.1 | 1,551 | 15.1 |
| 6+ | 420 | 14.3 | 267 | 7.8 | 171 | 4.3 | 857 | 8.3 |
| **Stunted** | | | | | | | | |
| Not stunted | 1,555 | 53.0 | 1,945 | 57.1 | 2,575 | 65.1 | 6,075 | 59.0 |
| Stunted | 1,305 | 44.5 | 1,387 | 40.7 | 1,311 | 33.1 | 4,003 | 38.9 |
| **Wasted** | | | | | | | | |
| Not wasted | 2,629 | 89.6 | 2,970 | 87.2 | 3,518 | 89.0 | 9,117 | 88.6 |
| Wasted | 232 | 7.9 | 362 | 10.6 | 368 | 9.3 | 961 | 9.3 |
| **Underweight** | | | | | | | | |

*(Continued)*

**Table 1.** (Continued)

| Variables | | 2005 (n = 2,933) | | 2010 (n = 3,407) | | 2014 (n = 3,955) | | 2005–2014 (n = 10,294) | |
|---|---|---|---|---|---|---|---|---|---|
| | | Freq. | % | Freq. | % | Freq. | % | Freq. | % |
| | Not Underweight | 1,997 | 68.1 | 2,366 | 69.4 | 2,892 | 73.1 | 7,256 | 70.5 |
| | Underweight | 863 | 29.4 | 965 | 28.3 | 994 | 25.1 | 2,822 | 27.4 |
| **Recent fever** | | | | | | | | | |
| | No | 1,860 | 63.4 | 2,383 | 69.9 | 2,774 | 70.1 | 7,017 | 68.2 |
| | Yes | 1,073 | 36.6 | 1,024 | 30.1 | 1,180 | 29.8 | 3,277 | 31.8 |
| **Recent diarrhea** | | | | | | | | | |
| | No | 2,357 | 80.4 | 2,856 | 83.8 | 3,413 | 86.3 | 8,626 | 83.8 |
| | Yes | 576 | 19.6 | 551 | 16.2 | 541 | 13.7 | 1,668 | 16.2 |
| **Vitamin A supplement** | | | | | | | | | |
| | No | 1,925 | 65.6 | 1,041 | 30.6 | 1,226 | 31.0 | 4,193 | 40.7 |
| | Yes | 1,008 | 34.4 | 2,366 | 69.4 | 2,729 | 69.0 | 6,102 | 59.3 |
| **Iron supplement** | | | | | | | | | |
| | No | 2,901 | 98.9 | 3,352 | 98.4 | 3,711 | 93.8 | 9,964 | 96.8 |
| | Yes | 32 | 1.1 | 55 | 1.6 | 244 | 6.2 | 331 | 3.2 |
| **Parasite medication** | | | | | | | | | |
| | No | 2,155 | 73.5 | 1,438 | 42.2 | 1,586 | 40.1 | 5,180 | 50.3 |
| | Yes | 778 | 26.5 | 1,969 | 57.8 | 2,363 | 59.7 | 5,109 | 49.6 |
| **Household Characteristics** | | | | | | | | | |
| **Place of residence** | | | | | | | | | |
| | Urban | 380 | 13.0 | 523 | 15.4 | 545 | 13.8 | 1,448 | 14.1 |
| | Rural | 2,553 | 87.0 | 2,884 | 84.6 | 3,409 | 86.2 | 8,846 | 85.9 |
| **Wealth index** | | | | | | | | | |
| | Poorest | 763 | 26.0 | 895 | 26.3 | 1,011 | 25.6 | 2,669 | 25.9 |
| | Poorer | 668 | 22.8 | 708 | 20.8 | 792 | 20.0 | 2,168 | 21.1 |
| | Middle | 551 | 18.8 | 639 | 18.8 | 764 | 19.3 | 1,955 | 19.0 |
| | Richer | 482 | 16.4 | 605 | 17.8 | 670 | 16.9 | 1,757 | 17.1 |
| | Richest | 469 | 16.0 | 560 | 16.4 | 717 | 18.1 | 1,746 | 17.0 |
| **Source of drinking water** | | | | | | | | | |
| | Improved | 1,537 | 52.4 | 1,839 | 54.0 | 2,006 | 50.7 | 5,381 | 52.3 |
| | Unimproved | 1,396 | 47.6 | 1,568 | 46.0 | 1,949 | 49.3 | 4,913 | 47.7 |
| **Type of toilet facility** | | | | | | | | | |
| | Improved | 573 | 19.5 | 1,177 | 34.5 | 1,934 | 48.9 | 3,684 | 35.8 |
| | Non-Improved | 2,360 | 80.5 | 2,230 | 65.5 | 2,021 | 51.1 | 6,610 | 64.2 |
| **Region** | | | | | | | | | |
| | Plain | 1,326 | 45.2 | 1,655 | 48.6 | 1,770 | 44.8 | 4,751 | 46.2 |
| | Tonle Sap | 998 | 34.0 | 1,066 | 31.3 | 1,264 | 32.0 | 3,328 | 32.3 |
| | Coastal | 215 | 7.3 | 210 | 6.2 | 236 | 6.0 | 661 | 6.4 |
| | Plateau/Mountain | 395 | 13.5 | 476 | 14.0 | 685 | 17.3 | 1556 | 15.1 |

Notes: Data Source: CDHS 2005, 2020, and 2014. Survey weights applied to obtain weighted percentages across survey years and pooled data.

mother who was anemic, 3.7% had a mother who smoked, 86% resided in rural areas, and 26% belonged to the poorest households (see **Table 1**). The overall prevalence of anemia among children decreased slightly from 62.2% in 2005 to 56.6% in 2014 (see **Fig 2**). The extent of this

downward trend varied across the four geographical regions; the highest prevalence of anemia in the Tonle Sap region in 2005 (69.5%). The coastal region had the lowest prevalence of anemia across all survey years, with prevalence levels in 2005 that were lower than the prevalence levels of other regions in 2014 (see **Fig 2**). Anemia was highest among children in Pursat (84.3%), in Kampong Thom province (67%), and in Preah Vihear and Steung Treng (68.6%) for 2005, 2010 and 2014, respectively (see **Fig 3** **and S2 Table**). While the lowest prevalence of anemia among children was observed in Kampot and Kep provinces (48.3%), in Pursat (39.7%), and in Phnom Penh (41.3%) for 2005, 2010 and 2014.

Preliminary analyses included evaluation of cross-tabular frequency distributions combined with Chi-square tests (see **Table 2**). Maternal age was significantly associated with the risk of anemia in children in survey years 2010, and 2014; the largest percentage of children with anemia were those with younger mothers aged less than 19 years (71.3% in 2005, 71.4% in 2010, and 71.6% in 2014). Maternal educational attainment was significantly associated with anemia in children in all survey years; the largest percentage of children with anemia were generally those whose mothers had no formal education (68.5% in 2005, 58.3% in 2010, and 57.2% in 2014). Maternal anemia was also significantly associated with anemia in children; the prevalence of anemia among children who had anemic mothers was 69% in 2005, 63.1% in 2010, and 64% in 2010.

Anemia was related to children's ages, with a higher prevalence in the age group of 6 to 11 months (6–11 months: 84.3% in 2005, 83.3% in 2010, and 80.6% in 2014; 12–23 months: 79.1%, 77.1%, and 73.7%, respectively). In addition, anemia among male children was higher than female children: 64.4% in 2005, 58% in 2010, and 58.1% in 2014. Stunted child had higher

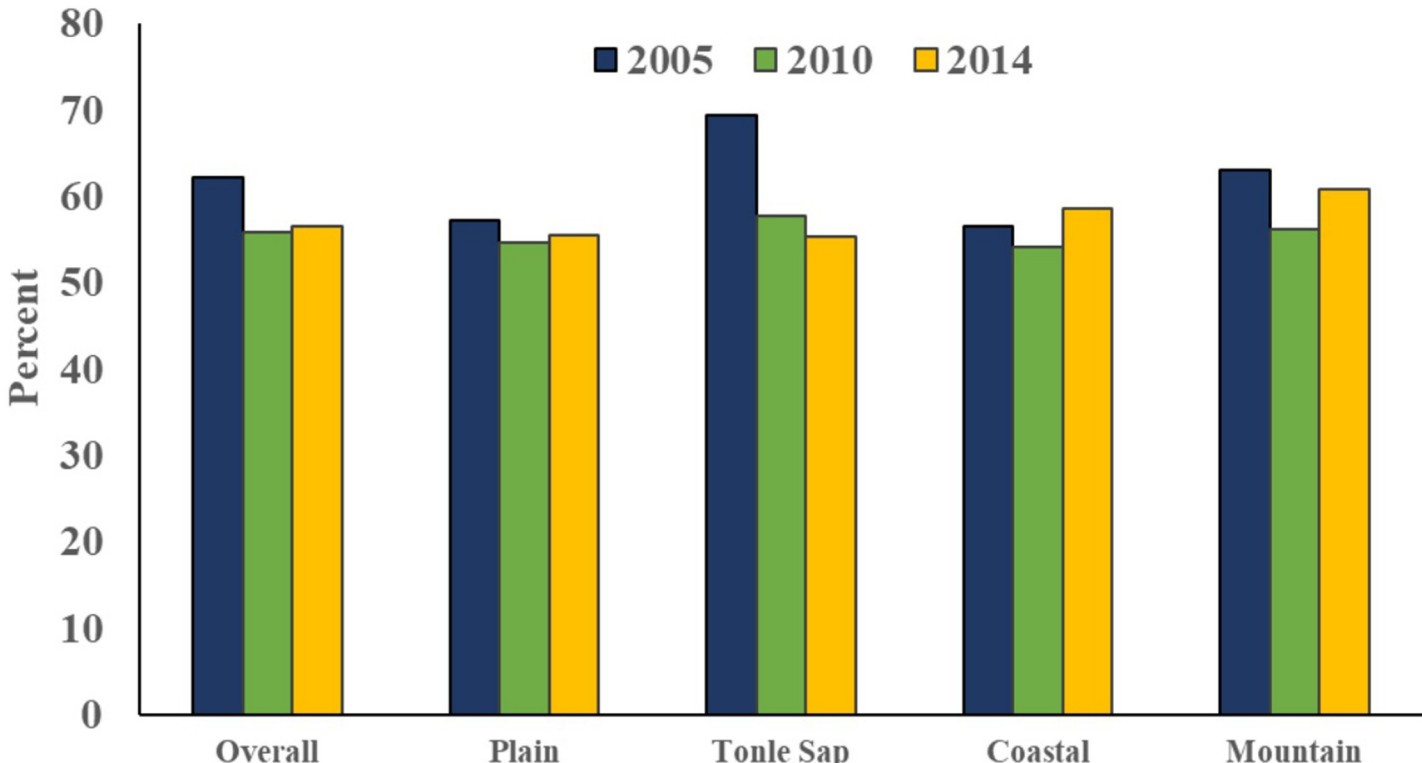

**Fig 2. Overall and regional trends of anemia among children age 6–59 months by survey year. Notes:** Data Source: CDHS 2005, 2020, and 2014. Survey weights applied to obtain weighted percentages across survey years and pooled data.

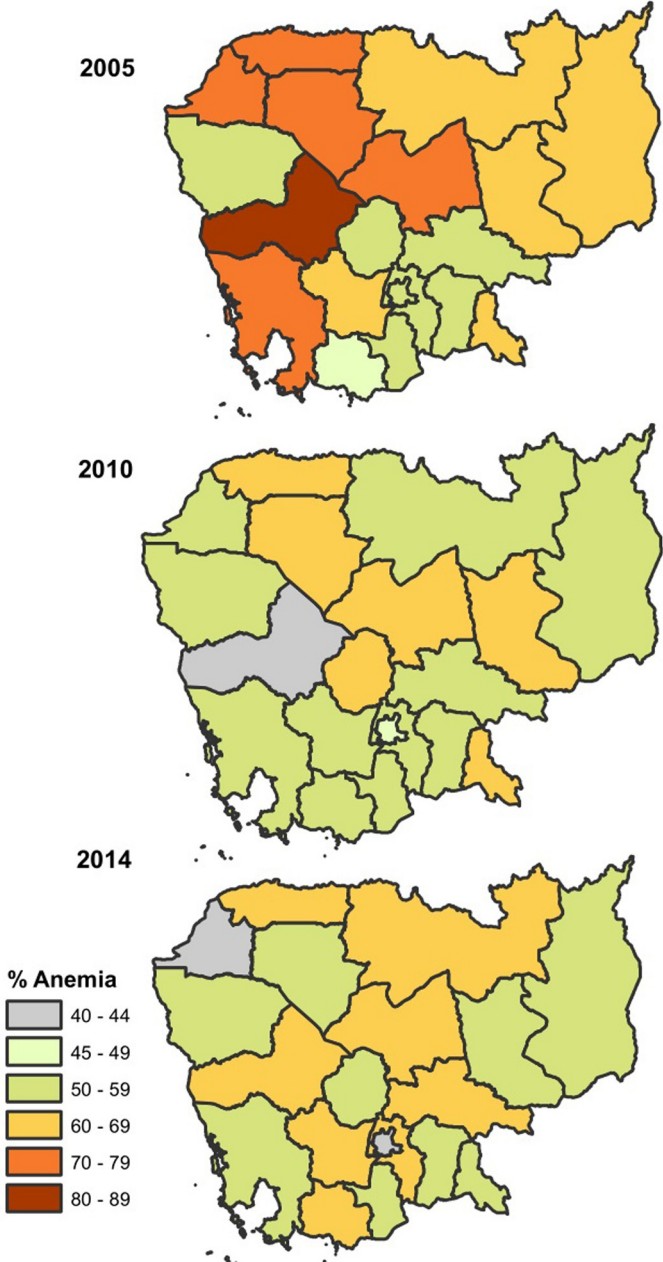

**Fig 3. Geographical distribution of anemia among children 6–59 months.** Shapefiles for administrative boundaries in Cambodia are publicly accessible through DHS website at (https://spatialdata.dhsprogram.com/boundaries/#view= table&countryId=KH). **Notes:** Survey weights applied to obtain weighted percentages across survey years and pooled data.

prevalence of anemia (65.3% in 2005, 57.2% in 2010 and 61.5% in 2014) as did children who were underweight (65% in 2005, 60% in 2010, and 61.3% in 2014). Having had fever or diarrhea in the last 2 weeks was associated with anemia among children in all of the survey years; among children with a history of fever, 67.2% in 2005, 61.1% in 2010, and 62.4% in 2014 were anemic; among children with diarrhea, 70.4% in 2005, 68.8% in 2010, and 66.4% in 2014 were anemic, respectively. Deworming for intestinal parasites was also associated with anemia in children. Results show that among children who had taken drugs for intestinal parasites in the

last 6 months, 55.3% in 2005, 52.1% in 2010, and 51.7% in 2014 were anemic. The proportion of anemic children in rural areas was higher than in urban areas in all three years, at 62.5% in 2005, 58% in 2010, and 59% in 2014. Anemia was also associated with household toilet facility; across the survey years, 64.5% in 2005, 58.5% in 2010 and 62.5% in 2014 of children in households that reported having no toilet facility were anemic. The proportion of children with anemia was highest among children in the poorest households across all the survey years: 69.4% in 2005, 60.4% in 2010 and 65% in 2014 were anemic, respectively.

After adjusting for other individual or household-level characteristics (see **Table 3**), children aged 6–59 months in Cambodia were more likely to have anemia if their mothers were likewise anemic (AOR = 1.77, 95% CI: 1.58–1.97) compared with children of non-anemic mothers; if they were male compared to female (AOR = 1.20, 95% CI: 1.07–1.33) ; underweight (95% CI: 1.14–1.57) or stunted (95% CI: 1.09–1.41) compared to children who were not underweight or stunted; if they had recently experienced a fever (AOR = 1.16, 95% CI: 1.03–1.31) or a bout of diarrhea (AOR = 1.16, 95% CI: 0.97–1.38) compared to children who had not recently experienced a fever or diarrhea; or if they were residing in rural areas (AOR = 1.10, 95% CI: 0.92–1.31) compared to children residing in urban areas. Children were less likely to have anemia if they were older, came from a wealthier household, or had medications for intestinal paraites in the last 6 months. With respect to age, children aged 12–23 months (AOR = 0.68; 95% CI: 0.55–0.86), 24–35 months (AOR = 0.21; 95% CI: 0.17–0.26), 36–47 months (AOR = 0.17; 95% CI: 0.13–0.21) and 48–59 months (AOR = 0.15; 95% CI: 0.12–0.19) were less likely to be anemic than those aged 6–11 months. Children from the wealthiest households had lower odds of being anemic compared with those in the poorest households (AOR = 0.66; 95% CI: 0.50–0.86). The odds of being anemic were lower by 14% (AOR = 0.86; 95% CI: 0.89–0.93) among children who took medications for intestinal parasites compared to those who did not take similar medications (see **Fig 4**).

To evaluate potential effect modification of statistically signficiant associations by time and space, interaction models were estimated for each of the statistically significant factors from the adjusted analysis presented in **Table 3**. These factors were separately interacted with region and year. Almost all of these analyses resulted in interaction terms that were not statistically significant, indicating that the corresponding factors were associated with child anemia regardless of the region of Cambodia where a child lived or the year in which the data were collected. The one exception was the interaction between parasite treament and region, which resulted in statistically significant interaction that was visualized with predicted probabilities (see **Fig 5**). The predicted probabilities indicate that treatment for parasites was protective in the mountainous and tonle sap regions of Cambodia as children who had recently received a treament in these regions had significantly lower probability of anemia compared to other children.

## Discussion

We have described the temporal and geospatial prevalence of anemia among children in Cambodia between 2005 and 2014 and noted only a slight decline during this period. The overall prevalence of anemia among children was 62.2% in 2005, 55.8% in 2010, and 56.6% in 2014, which is similar to the prevalence of anemia among children under five in South Asia (61.5% in 2005, 57.4% in 2010, 54.4% in 2014, and 52% in 2019) [1, 32]. According to WHO (2008), the cut-off values for public health significance are such that when the prevalence of anemia is 40% and above, the country is considered to have a serious public health problem [2]. Estimates from the WHO indicate that this pattern of minimal decline continued through 2020

**Table 2. Chi-square test results for associations between child anemia and child, maternal, and household characteristics.**

| Variables | 2005 (n = 2,933) | | | 2010 (n = 3,407) | | | 2014 (n = 3,955) | | | 2005–2014 (n = 10,294) | | |
|---|---|---|---|---|---|---|---|---|---|---|---|---|
| | Not Anemic | Anemic | p-value | Not Anemic | Anemic | p-value | Not Anemic | Anemic | p-value | Not Anemic | Anemic | p-value |
| **Maternal Characteristics** | | | | | | | | | | | | |
| **Age (years)** | | | | | | | | | | | | |
| 15–20 | 28.7 | 71.3 | 0.241 | 28.6 | 71.4 | <0.001 | 28.2 | 71.8 | 0.002 | 28.4 | 71.6 | <0.001 |
| 20–34 | 38.3 | 61.7 | | 44.0 | 56.0 | | 43.9 | 56.1 | | 42.5 | 57.5 | |
| 35–49 | 38.0 | 62.0 | | 48.7 | 51.3 | | 46.1 | 53.9 | | 43.8 | 56.2 | |
| **Education** | | | | | | | | | | | | |
| No education | 31.5 | 68.5 | <0.001 | 41.7 | 58.3 | <0.001 | 42.8 | 57.2 | <0.001 | 38.2 | 61.8 | <0.001 |
| Primary | 37.7 | 62.3 | | 43.4 | 56.6 | | 41.1 | 58.9 | | 40.8 | 59.2 | |
| Secondary plus | 47.6 | 52.4 | | 48.0 | 52.0 | | 47.9 | 52.1 | | 47.9 | 52.5 | |
| **Marital status** | | | | | | | | | | | | |
| Ever married | 42.7 | 57.3 | 0.294 | 38.2 | 61.8 | 0.201 | 48.6 | 51.4 | 0.239 | 43.5 | 56.5 | 0.592 |
| Married | 37.6 | 62.4 | | 44.5 | 55.5 | | 43.2 | 56.8 | | 42.0 | 58.0 | |
| **Employment** | | | | | | | | | | | | |
| Not working | 31.9 | 68.1 | 0.611 | 44.4 | 55.6 | 0.933 | 40.3 | 59.7 | <0.109 | 41.6 | 58.4 | 0.762 |
| Working | 37.8 | 62.2 | | 44.1 | 55.9 | | 44.5 | 55.5 | | 42.2 | 57.8 | |
| **Anemia status** | | | | | | | | | | | | |
| Non anemic | 44.8 | 55.2 | <0.001 | 36.9 | 50.0 | <0.001 | 49.6 | 50.4 | <0.001 | 48.4 | 51.6 | <0.001 |
| Anemic | 31.1 | 68.9 | | 50.0 | 63.1 | | 36.0 | 64.0 | | 34.8 | 65.2 | |
| **Smoking** | | | | | | | | | | | | |
| No-smoker | 38.1 | 61.9 | 0.266 | 44.5 | 55.5 | 0.102 | 43.8 | 56.2 | 0.011 | 42.5 | 57.5 | <0.001 |
| Smoker | 33.2 | 66.8 | | 33.9 | 66.1 | | 29.9 | 70.1 | | 32.4 | 67.6 | |
| **Child Characteristics** | | | | | | | | | | | | |
| **Place of birth** | | | | | | | | | | | | |
| Hospital | 40.9 | 59.1 | 0.196 | 43.8 | 56.2 | 0.766 | 44.1 | 59.9 | 0.087 | 43.7 | 56.3 | 0.005 |
| Home | 37.1 | 62.9 | | 44.5 | 55.5 | | 40.1 | 59.9 | | 40.1 | 59.9 | |
| **Age** | | | | | | | | | | | | |
| 6–11 | 15.7 | 84.3 | <0.001 | 16.7 | 83.3 | <0.001 | 19.4 | 80.6 | <0.001 | 17.5 | 82.5 | <0.001 |
| 12–23 | 20.9 | 79.1 | | 22.9 | 77.1 | | 26.3 | 73.7 | | 23.7 | 76.3 | |
| 24–35 | 42.3 | 57.7 | | 54.3 | 45.7 | | 49.4 | 50.6 | | 49.1 | 50.9 | |
| 36–47 | 48.7 | 51.3 | | 58.1 | 41.9 | | 54.1 | 45.9 | | 53.8 | 46.2 | |
| 48–59 | 51.9 | 48.1 | | 58.2 | 41.8 | | 59.5 | 40.5 | | 56.9 | 43.1 | |
| **Sex** | | | | | | | | | | | | |
| Female | 39.9 | 60.1 | 0.063 | 42.1 | 53.5 | 0.039 | 45.1 | 54.9 | 0.099 | 44.0 | 56.0 | 0.002 |
| Male | 35.6 | 64.4 | | 46.5 | 57.9 | | 41.9 | 58.1 | | 40.2 | 59.8 | |
| **Birth order** | | | | | | | | | | | | |
| 1–2 | 39.0 | 61.0 | 0.240 | 44.5 | 55.5 | 0.717 | 43.9 | 56.1 | 0.435 | 42.9 | 57.1 | 0.040 |
| 3–5 | 36.8 | 63.2 | | 42.0 | 58.0 | | 41.5 | 58.5 | | 39.8 | 60.2 | |
| 6+ | 33.8 | 66.2 | | 45.2 | 54.8 | | 38.9 | 61.1 | | 38.3 | 61.7 | |
| **Stunted** | | | | | | | | | | | | |
| Not stunted | 40.6 | 59.4 | 0.008 | 44.8 | 55.2 | 0.368 | 46.0 | 54.0 | <0.001 | 44.3 | 55.7 | <0.001 |
| Stunted | 34.7 | 65.3 | | 42.8 | 57.2 | | 38.5 | 61.5 | | 38.8 | 61.2 | |
| **Wasted** | | | | | | | | | | | | |
| Not wasted | 37.4 | 62.6 | 0.132 | 44.8 | 55.2 | 0.032 | 44.5 | 55.5 | 0.002 | 42.5 | 57.5 | 0.028 |
| Wasted | 43.9 | 56.1 | | 37.8 | 62.2 | | 34.2 | 65.8 | | 37.9 | 62.1 | |
| **Underweight** | | | | | | | | | | | | |
| Not Underweight | 39.1 | 60.9 | 0.087 | 45.5 | 54.5 | 0.041 | 45.2 | 54.8 | 0.003 | 43.6 | 56.4 | <0.001 |

*(Continued)*

**Table 2.** (Continued)

| Variables | 2005 (n = 2,933) | | | 2010 (n = 3,407) | | | 2014 (n = 3,955) | | | 2005–2014 (n = 10,294) | | |
|---|---|---|---|---|---|---|---|---|---|---|---|---|
| | Not Anemic | Anemic | p-value | Not Anemic | Anemic | p-value | Not Anemic | Anemic | p-value | Not Anemic | Anemic | p-value |
| Underweight | 35.2 | 64.8 | | 40.2 | 59.8 | | 38.7 | 61.3 | | 38.1 | 61.9 | |
| **Recent fever** | | | | | | | | | | | | |
| No | 40.7 | 59.3 | <0.001 | 46.4 | 53.6 | <0.001 | 45.9 | 54.1 | <0.001 | 44.7 | 55.3 | <0.001 |
| Yes | 32.8 | 67.2 | | 38.9 | 61.1 | | 37.6 | 62.4 | | 36.5 | 63.6 | |
| **Recent diarrhea** | | | | | | | | | | | | |
| No | 39.8 | 60.2 | <0.001 | 46.7 | 53.3 | <0.001 | 45.0 | 55.0 | <0.001 | 44.1 | 55.9 | <0.001 |
| Yes | 29.6 | 70.4 | | 31.2 | 68.8 | | 33.6 | 66.4 | | 31.4 | 68.6 | |
| **Vitamin A supplement** | | | | | | | | | | | | |
| No | 38.7 | 61.3 | 0.315 | 43.4 | 56.6 | 0.635 | 39.0 | 61.0 | 0.007 | 39.9 | 60.1 | 0.007 |
| Yes | 36.2 | 63.8 | | 44.5 | 55.5 | | 45.4 | 54.6 | | 43.5 | 56.5 | |
| **Iron supplement** | | | | | | | | | | | | |
| No | 37.9 | 67.6 | 0.577 | 44.1 | 49.8 | 0.464 | 43.7 | 60.8 | 0.280 | 42.1 | 59.6 | 0.614 |
| Yes | 32.4 | 62.1 | | 50.2 | 55.9 | | 39.2 | 56.3 | | 40.4 | 57.9 | |
| **Parasite medication** | | | | | | | | | | | | |
| No | 35.3 | 64.7 | <0.001 | 39.2 | 60.8 | <0.001 | 36.3 | 63.7 | <0.001 | 36.7 | 63.3 | <0.001 |
| Yes | 44.7 | **55.3** | | 47.9 | **52.1** | | 48.3 | **51.7** | | 47.6 | **52.5** | |
| **Household Characteristics** | | | | | | | | | | | | |
| **Place of residence** | | | | | | | | | | | | |
| Urban | 39.8 | 60.2 | 0.608 | 54.8 | 45.2 | <0.001 | 56.6 | 43.4 | <0.001 | 51.5 | 48.5 | <0.001 |
| Rural | 37.5 | 62.5 | | 42.3 | 57.7 | | 41.3 | 58.7 | | 40.5 | 59.5 | |
| **Wealth index** | | | | | | | | | | | | |
| Poorest | 30.6 | 69.4 | <0.001 | 39.6 | 60.4 | <0.001 | 35.0 | 65.0 | <0.001 | 35.3 | 64.7 | <0.001 |
| Poorer | 34.9 | 65.1 | | 41.4 | 58.6 | | 38.5 | 61.5 | | 38.3 | 61.7 | |
| Middle | 38.9 | 61.1 | | 41.8 | 58.2 | | 45.4 | 54.6 | | 42.4 | 57.6 | |
| Richer | 42.0 | 58.0 | | 45.4 | 54.6 | | 45.9 | 54.1 | | 44.7 | 55.3 | |
| Richest | 48.1 | 51.9 | | 56.4 | 43.6 | | 56.4 | 43.6 | | 54.2 | 45.8 | |
| **Source of drinking water** | | | | | | | | | | | | |
| Improved | 38.1 | 61.9 | 0.763 | 45.4 | 54.6 | 0.265 | 44.0 | 56.0 | 0.556 | 42.8 | 57.2 | 0.235 |
| Unimproved | 37.4 | 62.6 | | 42.8 | 57.2 | | 42.8 | 57.2 | | 41.3 | 58.7 | |
| **Type of toilet facility** | | | | | | | | | | | | |
| Improved | 47.4 | 52.6 | <0.001 | 49.2 | 50.8 | <0.001 | 49.8 | 50.2 | <0.001 | 49.2 | 50.8 | <0.001 |
| Non-Improved | 35.5 | 64.5 | | 41.5 | 58.5 | | 37.4 | 62.6 | | 38.1 | 61.9 | |
| **Region** | | | | | | | | | | | | |
| Plain | 42.7 | 57.3 | <0.001 | 45.3 | 54.7 | 0.572 | 44.4 | 55.6 | 0.223 | 44.3 | 55.7 | 0.004 |
| Tonle Sap | 30.5 | 69.5 | | 42.3 | 57.7 | | 44.7 | 55.3 | | 39.7 | 60.3 | |
| Coastal | 43.4 | 56.6 | | 45.8 | 54.2 | | 41.3 | 58.7 | | 43.4 | 56.6 | |
| Mountain | 36.9 | 63.1 | | 43.7 | 56.3 | | 39.2 | 60.8 | | 40.0 | 60.0 | |

Notes: Data Source: CDHS 2005, 2020, and 2014. Survey weights applied to obtain weighted percentages across survey years and pooled data.

and child anemia may have increased more recently due to challenges such as increased food insecurity associated with the COVID-19 pandemic [5, 33].

This study suggests that Cambodia did not achieve its national nutritional targets aimed at reducing the prevalence of anemia among children down to 42% in 2015. This lapse occurred

**Table 3. Unadjusted and adjusted analyses of child anaemia in Cambodia.**

| Variables | | Unadjusted (N = 10,294) | | Adjusted (N = 10,051) | |
|---|---|---|---|---|---|
| | | OR | 95%CI | AOR | 95%CI |
| **Mothers age (years)** | | | | | |
| | 15–20 | Ref. | Ref. | Ref. | Ref. |
| | 20–34 | 0.54*** | (0.41–0.70) | 0.92 | (0.68–1.25) |
| | 35–49 | 0.51*** | (0.38–0.68) | 0.87 | (0.62–1.21) |
| **Maternal education** | | | | | |
| | No education | Ref. | Ref. | Ref. | Ref. |
| | Primary | 0.89 | (0.79–1.02) | 1.00 | (0.87–1.16) |
| | Secondary and Higher | 0.67*** | (0.58–0.78) | 0.97 | (0.80–1.17) |
| **Maternal marital status** | | | | | |
| | Ever married | Ref. | Ref. | | |
| | Married | 1.06 | (0.85–1.33) | | |
| **Maternal employment** | | | | | |
| | Not working | Ref. | Ref. | | |
| | Working | 0.98 | (0.84–1.14) | | |
| **Maternal anemia** | | | | | |
| | Non anemic | Ref. | Ref. | 1.00 | (1.00–1.00) |
| | Anemic | 1.76*** | (1.59–1.95) | **1.76***** | **(1.58–1.97)** |
| **Maternal smoking** | | | | | |
| | No-smoker | Ref. | Ref. | Ref. | Ref. |
| | Smoker | 1.54** | (1.19–1.99) | 1.22 | (0.91–1.63) |
| **Place of birth** | | | | | |
| | Hospital | Ref. | Ref. | Ref. | Ref. |
| | Home | 1.16** | (1.05–1.28) | 0.95 | (0.84–1.10) |
| **Child age** | | | | | |
| | 6–11 | Ref. | Ref. | | |
| | 12–23 | 0.68** | (0.55–0.85) | **0.68***** | **(0.54–0.86)** |
| | 24–35 | 0.22*** | (0.18–0.27) | **0.21***** | **(0.17–0.26)** |
| | 36–47 | 0.18*** | (0.15–0.22) | **0.17***** | **(0.13–0.21)** |
| | 48–59 | 0.16*** | (0.13–0.20) | **0.15***** | **(0.12–0.20)** |
| **Child sex** | | | | | |
| | Female | Ref. | Ref. | Ref. | Ref. |
| | Male | 1.17** | (1.06–1.29) | **1.20***** | **(1.07–1.33)** |
| **Birth order** | | | | | |
| | 1–2 | Ref. | Ref. | Ref. | Ref. |
| | 3–5 | 1.14* | (0.99–1.31) | 1.07 | (0.91–1.27) |
| | 6+ | 1.21* | (1.00–1.47) | 1.09 | (0.85–1.39) |
| **Child stunted** | | | | | |
| | Not stunted | Ref. | Ref. | Ref. | Ref. |
| | Stunted | 1.25*** | (1.13–1.39) | **1.24***** | **(1.09–1.41)** |
| **Child wasted** | | | | | |
| | Not wasted | Ref. | Ref. | Ref. | Ref. |
| | Wasted | 1.21* | (1.02–1.44) | 0.92 | (0.75–1.12) |
| **Child underweight** | | | | | |
| | Not underweight | Ref. | Ref. | Ref. | Ref. |
| | Underweight | 1.25*** | (1.12–1.40) | **1.27***** | **(1.10–1.47)** |
| **Recent fever** | | | | | |

*(Continued)*

**Table 3.** (Continued)

| Variables | | Unadjusted (N = 10,294) | | Adjusted (N = 10,051) | |
|---|---|---|---|---|---|
| | | OR | 95%CI | AOR | 95%CI |
| | No | Ref. | Ref. | Ref. | Ref. |
| | Yes | 1.41*** | (1.27–1.57) | 1.16* | (1.03–1.31) |
| **Recent diarrhea** | | | | | |
| | No | Ref. | Ref. | Ref. | Ref. |
| | Yes | 1.72*** | (1.48–2.00) | 1.16 | (0.97–1.38) |
| **Vitamin A supplement** | | | | | |
| | No | Ref. | Ref. | Ref. | Ref. |
| | Yes | 0.86** | (0.77–0.96) | 1.04 | (0.91–1.20) |
| **Taken iron** | | | | | |
| | No | Ref. | Ref. | Ref. | Ref. |
| | Yes | 1.08 | (0.81–1.43) | 1.30 | (0.94–1.78) |
| **Parasite medication** | | | | | |
| | No | Ref. | Ref. | Ref. | Ref. |
| | Yes | 0.64*** | (0.58–0.71) | **0.86*** | **(0.76–0.98)** |
| **Place of residence** | | | | | |
| | Urban | Ref. | Ref. | Ref. | Ref. |
| | Rural | 1.56*** | (1.35–1.81) | 1.10 | (0.93–1.32) |
| **Wealth index** | | | | | |
| | Poorest | Ref. | Ref. | Ref. | Ref. |
| | Poorer | 0.88* | (0.76–1.01) | 0.95 | (0.81–1.11) |
| | Middle | 0.74*** | (0.63–0.87) | 0.84 | (0.70–1.01) |
| | Richer | 0.68*** | (0.58–0.79) | 0.85 | (0.68–1.06) |
| | Richest | 0.46*** | (0.39–0.54) | **0.66**** | **(0.50–0.86)** |
| **Source of drinking water** | | | | | |
| | Improved | Ref. | Ref. | | |
| | Unimproved | 1.06 | (0.96–1.18) | | |
| **Type of toilet facility** | | | | | |
| | Improved | Ref. | Ref. | Ref. | Ref. |
| | Non-improved | 1.57*** | (1.42–1.74) | 1.17 | (0.99–1.39) |
| **Region** | | | | | |
| | Plains | Ref. | Ref. | Ref. | Ref. |
| | Tonle Sap | 1.21** | (1.06–1.37) | 1.10 | (0.96–1.26) |
| | Coastal | 1.03 | (0.86–1.24) | 1.05 | (0.86–1.27) |
| | Plateau/Mountain | 1.19* | (1.03–1.37) | 1.01 | (0.86–1.19) |
| **Survey years** | | | | | |
| | 2005 | Ref. | Ref. | Ref. | Ref. |
| | 2010 | 0.77** | (0.67–0.88) | **0.85*** | **(0.72–0.99)** |
| | 2014 | 0.79** | (0.69–0.91) | 0.90 | (0.76–1.07) |

Notes: Data Source: CDHS 2005, 2020, and 2014. Survey weights applied to obtain weighted percentages across year and pooled data.

* $p < 0.05$

** $p < 0.01$

*** $p < 0.001$

despite having a national nutrition policy, strategy, and plan of action since 2009 and corresponding public health interventions implemented to support reductions in child anemia such

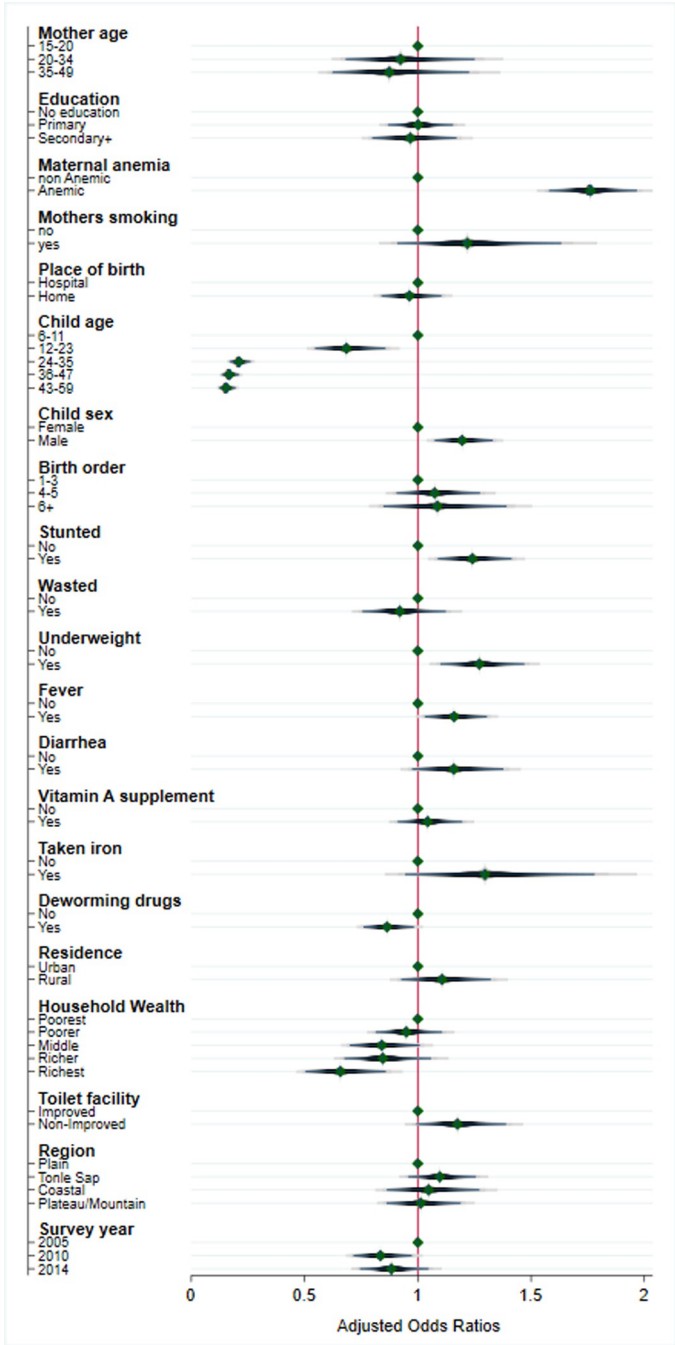

**Fig 4. Adjusted analyses of the odds of a child aged 6–59 months being anaemic in Cambodia (N = 10,051).**

as iron and folic acid supplements, complementary feeding, deworming, and breastfeeding [34]. This may be due, at least in part, to limited implementation and enforcement of these policies and interventions in remote areas of the country [34].

Lack of nationwide interventions to address the problem of anemia among Cambodian children was a major public health challenge. Data from the 2005–2014 Cambodia DHS indicated that the percentage of children 6–59 months who received deworming medication was

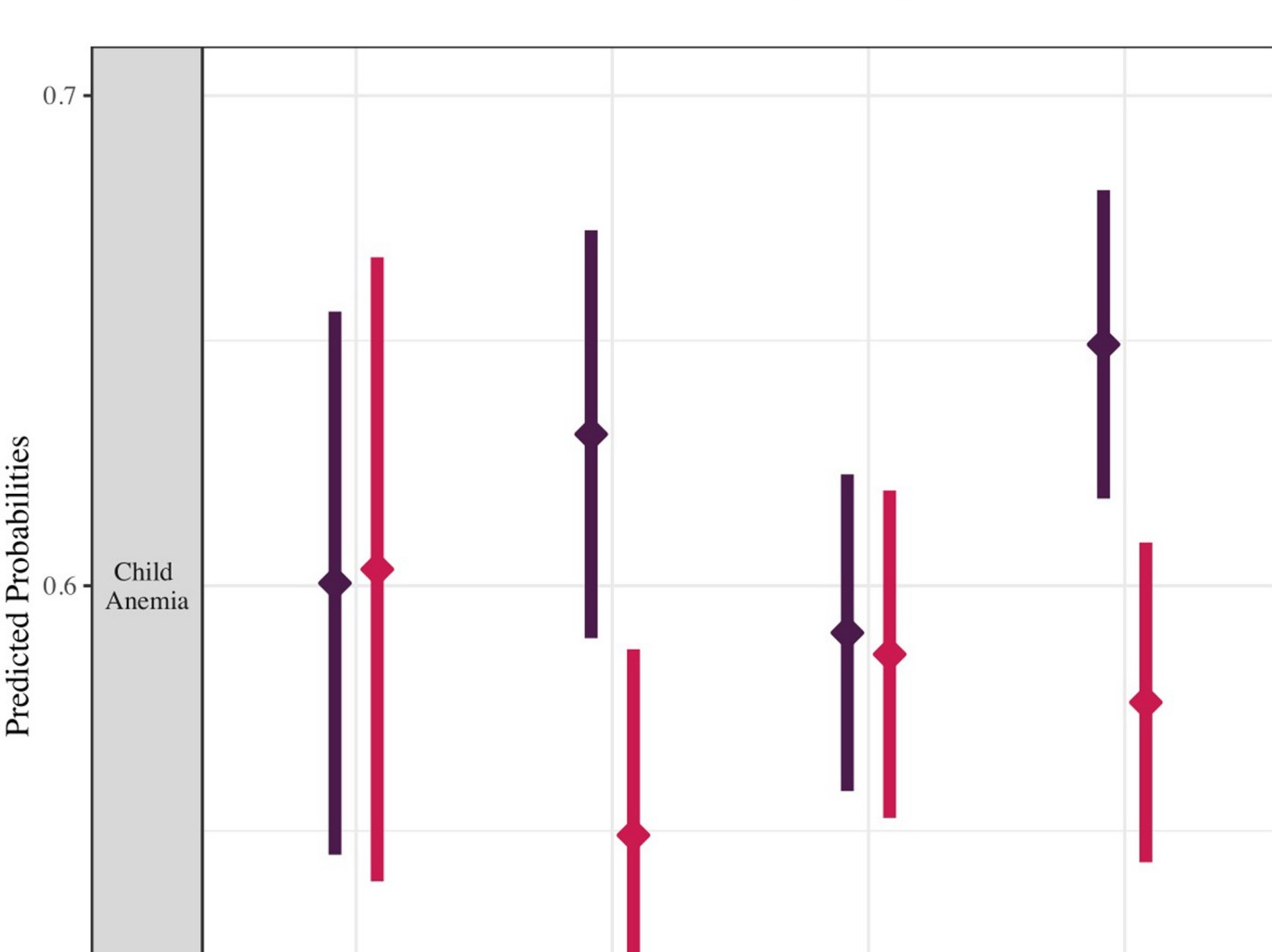

**Fig 5. Predicted probability of child anemia in Cambodia by region and parasitic treatment (N = 10,051).**

26.5% in 2005, 56.7% in 2010, 58.7% in 2014 [21–23]. Children were less likely to receive these medications if they resided in rural areas, were from poorer households, and had mothers with limited education [21–23]. A little over 2% of children took standard iron and folic acid supplements through the national health system in 2010 and this rose to 6% in 2014 [22, 23]. The

percentage of children who reported taking Vitamin A supplementation in last 6 months increased dramatically from 34.5% in 2005 to 70% in 2014; lower prevalence of supplementation was reported for children aged 6–8 months (39%) and 9–11 months (59%) in 2014 [22, 23]. In 2014, few children 6–23 months old were fed according to three key infant and young child feeding (IYCF) practices; e.g., early initiation (within one hour of birth) of exclusive breastfeeding, exclusive breastfeeding for the first six months of life, followed by nutritionally adequate and safe complementary foods [23]. In the last 10 years, Cambodia also piloted the Lucky Iron Fish Project to prevent anemia; however, despite evidence of a positive impact on iron status, factors such as cost and need for behavioral change have hampered large scale implementation [35]. Finally, intervention reported above also included markers of iron deficiency and the prevalence of iron deficiency and/or IDA can therefore be calculated [36]; hence the interest in better understanding associations between anemia and demographic, economic, and social factors as well as health-seeking behaviors across different parts of the country [37].

Our study found several factors associated with increased odds of having anemia among children aged 6–59 months in Cambodia, including maternal anemia, being male, being stunted, being underweight, and having a history of fever in the last 2 weeks. However, children living in the richest wealth quintiles and children aged 12–59 months were less likely to have anemia.

Maternal anemia increased the odds of having anemia among children. This finding is consistent with previous studies in Laos [14], South and Southeast Asia [15], Southern Africa [16], and Uganda [38]. Mothers and children often share the same socioeconomic environment and therefore the child's diet quality is likely similar to that of the mother [14, 15, 18, 39]. Women who live in low-income families are likely to have greater difficulty in purchasing and providing an iron-rich diet and other micro-nutrient-rich food for themselves and their children, which can increase the risk of developing childhood anemia [8, 15, 38]. However, this association was significant even after controlling for household wealth status, suggesting additional pathways beyond socioeconomic status. One potential pathway for newborns is that maternal anemia during pregnancy may contribute to low birth weight and preterm birth, both of which increase the risk of childhood anemia [40]. For infants, maternal anemia may also reduce iron content in breast milk and result in children having lower iron intake if exclusively breastfed while their mothers are anemic [41, 42].

Children aged 12–59 months were less likely to have anemia as compared to those of 6–11 months. This is consistent with other studies showing that anemia was more prevalent among children aged 6–11 months, which may stem from a higher need for iron during a child's early-life due to rapid growth and development [43], need for complementary feeding [44], and a heightened risk of diarrhea [45] and acute respiratory infection [46]. Male children have increased odds of having anemia. Similar results were found in studies in Laos [14] and sub-Saharan Africa [47]. This may be explained by physiological variations in gender or due to longitudinal and faster growth rates in boys than in girls [14, 47], which requires a higher demand for iron for their bodies [48], particularly during the early years of life [49].

In addition, stunted and underweight children had increased odds of having anemia. Malnutrition is a major public health challenge in Cambodia. A prevalence of stunted youth decreased from 32% to 22%, and those were underweight from 25% to 16.3%, among children under five years of age in 2014 and 2021–22 [50]. Being stunted and underweight is associated with acute and chronic malnutrition, which share common causes with anemia [51], and these factors are aggravated by food insecurity and poverty. Food insecurity and inadequate consumption of micronutrients such as iron, vitamin B12, and folate affect the nutritional status of children, which contributes to the development of anemia [51]. Therefore, public health

interventions should be targeted at improving the nutritional status of children by introducing diverse complementary foods enriched with micronutrients such as iron, vitamin B12, and folate from infancy.

Children with a history of fever had increased odds of having anemia, which is consistent with the results in previous studies conducted in Laos [14], South and Southeast Asia [15], and Southern Africa [16]. Fever among under-five children in Cambodia was a common symptom of acute and chronic inflammatory disease [34]. Cambodia is one of the malaria-endemic areas in Southeast Asia [52]. An association between childhood anemia and inflammatory infections, which includes fever, cough, and worm infestation, has been well established [53]. The possible pathway of this association is due to iron losses during the time of infections [54]. One previous study reported that iron-deficiency anemia may depend on inadequate iron-rich food intake, increased iron use in the body, and iron depletion due to parasitic infection [53]. Also, previous studies have reported that malaria increases erythrocyte destruction with a simultaneous failure of the bone marrow thus jeopardizing the ability to compensate for the losses [54].

Children from wealthier households were less likely to have anemia compared to children from poorer households. This association is consistent with previous studies [14, 15, 18, 39] and may reflect increased household access to resources such as nutrition and health care [38]. Better household income allows for living in good social and economic conditions, and hence it is easy for higher income families to provide a diverse diet, adequate care, and health services to their children [16]. Children who took drugs for intestinal parasites in previous 6 months had lower odds of having anemia compared to children who did not take drugs for intestinal parasites. It is consistent with studies in Thailand [55] and sub-Saharan Africa [56]. This may be due to the fact that intestinal parasites can induce anemia, and treating intestinal parasites with medication can reduce the incidence of anemia in children [57].

This study was conducted using pooled data from Cambodia Demographic and Health surveys from 2005, 2010, and 2014. This enabled us to describe the temporal and geographical trends in anemia among children aged 6–59 months using nationally representative data. Likewise, we were able to examine factors associated with anemia among children throughout Cambodia. However, as a cross-sectional study it was not possible to examine causality in these relationships. This study relied on hemoglobin as the measure of anemia based on WHO criteria; however, future studies could consider alternative red blood cell indices [58]. As an observational study that drew heavily upon self-reported data it is possible that there was some recall bias or social desirability bias in the data reported. As a secondary data analysis, several known risk factors for anemia, including parasitic infections such as malaria, and dietary information were not included in this study due to data limitations. The HemoCue device has been used widely in field settings where resources are limited since it is portable, simple to use, and reasonably priced; however, air bubbles, too much blood on the cuvettes' back side, over-filling of the cuvettes, and inadequate mixing of the samples can all provide inaccurate results [59]. Additionally, there is variability in Hemoglobin (Hb) and measurements dependent on Hemocue models, which may have affected the prevalence of anemia presented in this study [59]. Finally, the most recent round of CDHS data with information on anemia status was gathered in 2014 (Hemoglobin measurements were not collected in CDHS 2022). We acknowledge that given the span of time since the data used in this study were collected, the results from this study may no longer reflect present prevalence of anemia among children in Cambodia. However, estimates of anemia among children in Cambodia from the WHO suggest a pattern of minimal decline consistent with our findings that continued through 2020. Moreover, child anemia may have increased more recently due to challenges such as increased food insecurity associated with the COVID-19 pandemic [5, 33]. Future research should

further evaluate these relationships and the general prevalence of anemia among young Cambodian children aged 6–59 months once additional data become available.

## Conclusion

Our findings suggest that the high prevalence of anemia among children aged 6–59 months in Cambodia declined only slightly over the 10-year period between 2005 and 2014. Anemia among children continues to be a major public health concern in Cambodia. This study indicated that childhood anemia was associated with various factors, including being male, being younger, being stunted, being underweight, having a history of fever or diarrhea in the last 2 weeks, maternal anemia, being a member of the poorest families, did not take drugs for intestinal parasites and having a place of residence in a rural setting. Public health practitioners and policy makers should consider demographic groups represented by these characteristics to produce better targeted interventions to address anemia among children in Cambodia.

## Supporting information

**S1 Table. Results of checking multicollinearity using variance inflation factor.**
(DOCX)

**S2 Table. Prevalence of anemia in the 19 domains in CDHS 2005 2010 and 2014.**
(DOCX)

## Acknowledgments

The authors are grateful to DHS-ICF for the gathering, cleaning, and provision of the data used for this paper. We are indebted to Dr. **Chhea Chhorvann** and Dr. **Heng Sopheab** from the National Institute of Public Health, Phnom Penh, Cambodia for their encouragement, feedback, and support.

## Author Contributions

**Conceptualization:** Samnang Um.

**Formal analysis:** Samnang Um, Jonathan A. Muir.

**Methodology:** Samnang Um, Jonathan A. Muir.

**Project administration:** Samnang Um.

**Supervision:** Jonathan A. Muir.

**Validation:** Jonathan A. Muir.

**Visualization:** Samnang Um, Jonathan A. Muir.

**Writing – original draft:** Samnang Um.

**Writing – review & editing:** Samnang Um, Michael R. Cope, Jonathan A. Muir.

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
