## [Decision Letter · Decision Letter 0]

9 Jun 2023

PGPH-D-23-00986

Child Anemia in Cambodia: A Descriptive Analysis of Temporal and Geospatial Trends and Logistic Regression-Based Examination of Factors Associated with Anemia in Children Age 6-59 Months

Dear Dr. Um,

Thank you for submitting your manuscript to PLOS Global Public Health. After careful consideration, we feel that it has merit but does not fully meet PLOS Global Public Health’s publication criteria as it currently stands. Therefore, we invite you to submit a revised version of the manuscript that addresses the points raised during the review process.

We look forward to receiving your revised manuscript.

Kind regards,

Rajesh Sharma, Ph.D.

Academic Editor

Journal Requirements:

1. Please ensure that the Title in your manuscript and the Title in your online submission form are the same.

2. Please provide separate main figure files in .tif or .eps format only and ensure that all files are under our size limit of 10MB.

3. We noticed that you have two Table 2's in your manuscript. Please update your table numbers and cite them accordingly.

4. We notice that your supplementary tables are included in the manuscript file. Please remove them and upload them with the file type 'Supporting Information'. Please ensure that each Supporting Information file has a legend listed in the manuscript after the references list.

5. Some material included in your submission may be copyrighted. According to PLOS’s copyright policy, authors who use figures or other material (e.g., graphics, clipart, maps) from another author or copyright holder must demonstrate or obtain permission to publish this material under the Creative Commons Attribution 4.0 International (CC BY 4.0) License used by PLOS journals. Please closely review the details of PLOS’s copyright requirements here: PLOS Licenses and Copyright. If you need to request permissions from a copyright holder, you may use PLOS's Copyright Content Permission form.

Potential Copyright Issues:

Figure 2: please (a) provide a direct link to the base layer of the map (i.e., the country or region border shape) and ensure this is also included in the figure legend; and (b) provide a link to the terms of use / license information for the base layer image or shapefile. We cannot publish proprietary or copyrighted maps (e.g. Google Maps, Mapquest) and the terms of use for your map base layer must be compatible with our CC-BY 4.0 license. 

Additional Editor Comments (if provided):

Reviewers' comments:

Reviewer's Responses to Questions

**Comments to the Author**

1. Does this manuscript meet PLOS Global Public Health’s publication criteria? Is the manuscript technically sound, and do the data support the conclusions? The manuscript must describe methodologically and ethically rigorous research with conclusions that are appropriately drawn based on the data presented.

Reviewer #1: Yes

Reviewer #2: Yes

Reviewer #3: Yes

2. Has the statistical analysis been performed appropriately and rigorously?

Reviewer #1: Yes

Reviewer #2: Yes

Reviewer #3: Yes

3. Have the authors made all data underlying the findings in their manuscript fully available (please refer to the Data Availability Statement at the start of the manuscript PDF file)?

Reviewer #1: Yes

Reviewer #2: Yes

Reviewer #3: Yes

4. Is the manuscript presented in an intelligible fashion and written in standard English?

Reviewer #1: Yes

Reviewer #2: Yes

Reviewer #3: Yes

5. Review Comments to the Author

Reviewer #1: Title needs modification. It needs to be shorter, easily understandable and coherent to the objective of the study.

Don't forget to justify why you haven't used the data if there is any Demographic and Health Survey undertaken after 2014 in the country.

There are scholars and organizations which classify those with mild anemia as 'not anemic'. Consider if this apply in your case.

Better to mention if there was any data quality control measures undertaken especially Hemoglobin adjustments.

Recommendations has to be specific per the findings. If there are interventions being underway scaling it up and strengthening it also has to be considered.

Reviewer #2: Comments

1.Line 122-124 how the researchers arrived at the 10,434 is not clear. Further explanation on this will be helpful.

2.The summation of the frequencies (computation) in Table 1 under the Characteristics of Place of birth under section Home does not add up. Authors are encouraged to check these figures correctly.

3.Furthermore, the authors failed to perform dietary assessment on the mothers and again whether the babies where exclusively fed.

4.In my candid opinion based on the multiple limitation I suggest the title is modified slightly to represent the findings given that data was primarily a self-reported type.

5.The pre-maternal weights of the mothers could have been captured since these are normally in their prenatal books/records.

6.While the findings are interesting, I am not sure the findings represent the current situation given that the data were collected almost a decade ago. Will it be relevant today and what is the current situation of children anemia in Cambodia today?

Reviewer #3: Line 69

Which period? Please specify the period and reference

Line 136-140

Can SINGLE, WIDOWED and SEPARATED women be considered married?

In reality their social status is "NOT MARRIED" as they are not married.

Same comments with OCCUPATION

It would be good to make the following groups:

1. NOT WORKING (NO INCOME)

2. WORKING (others who work and earn money for family needs, so they have INCOME)

Line 145-146

Place of birth should be “HOSPITAL" and "HOME"

Being born in a public or private hospital is the same government objective (birth assisted by qualified personnel).

Indicators monitor are “Delivery assisted by qualified health worker” and “Delivery unassisted by qualified worker”

Line 148-150

It would be more comprehensive to specify here the period of recent fever and recent diarrhea.

Same comments to Vitamin A supplementation and intestinal parasite medication

RECENT is not clear

Line 259-262

The variables diarrhea, Fever and Place of residence are not significantly associated to Anemia in multivariate analysis because the confidence interval include “1” (not risk factor)

Line 287

It should be Table 3 instead Table 2

Line 306-308

This study presents the prevalence in 2005-2010 and 2015 whose results cannot be compared with the 2015 and 2020 objectives.

Instead, the results for 2015 and 2020 should be assessed against the country's objectives for these 2 years.

Line 311-312

The country has action plan since 2008. It should mean the country has developed or identified some intervention to address the problem.

If interventions are not implemented in the country to prevent anemia among children, it should be explained by some factors to be highlighted here. If the political commitment does not support strategies, the problem of anemia will always remain a health public health concern in Cambodia.

Line 314

Are you referring to univariate or multivariate analysis?

It's important to specify because recent diarrhea is not significantly associated with anemia in multivariate analysis

Line 319

The mother's anemia used in this analysis was assessed at what point in the life of the child included in the study?

If it was during the pregnancy of the child evaluated in the survey, the results of this analysis on the link between the child's anemia and that of the mother can be justified, relevant and explained.

On the other hand, if the mother's anemia occurred when the child was already at least 6 months old, I think there could be a bias, otherwise the analysis would have to be taken further, because the link between the child's anemia and the mother's anemia after the child's birth (because the target is children aged 6-59 months) raises questions, unless the mother has a chronic hematological disease.

Line 333

In addition, poorly conducted weaning in children aged 6 months and over is a cause of diarrhoeal disease and malnutrition (etc...), a major impact of which could be anemia in children due to nutritional imbalance or insufficient intake of micronutrients, including iron, vitamin B12…..

6. PLOS authors have the option to publish the peer review history of their article (what does this mean?). If published, this will include your full peer review and any attached files.

**Do you want your identity to be public for this peer review?** For information about this choice, including consent withdrawal, please see our Privacy Policy.

Reviewer #1: **Yes: **Shalama Lekasa Nagari

Reviewer #2: No

Reviewer #3: **Yes: **Kouadio Kobahounde, Epidemiologist

---

## [Decision Letter · Decision Letter 1]

17 Aug 2023

Child Anemia in Cambodia: A Descriptive Analysis of Temporal and Geospatial Trends and Logistic Regression-Based Examination of Factors Associated with Anemia in Children

PGPH-D-23-00986R1

Dear Mr. Um,

We are pleased to inform you that your manuscript 'Child Anemia in Cambodia: A Descriptive Analysis of Temporal and Geospatial Trends and Logistic Regression-Based Examination of Factors Associated with Anemia in Children' has been provisionally accepted for publication in PLOS Global Public Health.

Best regards,

Rajesh Sharma, Ph.D.

Academic Editor

Accept.

Reviewer Comments (if any, and for reference):

Reviewer's Responses to Questions

**Comments to the Author**

1. If the authors have adequately addressed your comments raised in a previous round of review and you feel that this manuscript is now acceptable for publication, you may indicate that here to bypass the “Comments to the Author” section, enter your conflict of interest statement in the “Confidential to Editor” section, and submit your "Accept" recommendation.

Reviewer #3: All comments have been addressed

2. Does this manuscript meet PLOS Global Public Health’s publication criteria? Is the manuscript technically sound, and do the data support the conclusions? The manuscript must describe methodologically and ethically rigorous research with conclusions that are appropriately drawn based on the data presented.

Reviewer #3: Yes

3. Has the statistical analysis been performed appropriately and rigorously?

Reviewer #3: Yes

4. Have the authors made all data underlying the findings in their manuscript fully available (please refer to the Data Availability Statement at the start of the manuscript PDF file)?

Reviewer #3: Yes

5. Is the manuscript presented in an intelligible fashion and written in standard English?

Reviewer #3: Yes

6. Review Comments to the Author

Reviewer #3: I took my time to carefully read the revised document. I confirm that the author has responded to my review comments. But I would be more comfortable if the age range was kept in the title (6-59 months). Indeed, the children are the age group of 0 to 17 years, and this study only analyzes children from 6 to 59 months.

7. PLOS authors have the option to publish the peer review history of their article (what does this mean?). If published, this will include your full peer review and any attached files.

**Do you want your identity to be public for this peer review?** For information about this choice, including consent withdrawal, please see our Privacy Policy.

Reviewer #3: **Yes: **KOUADIO KOBAHOUNDE
